# Stability and Catalase-Like Activity of a Mononuclear Non-Heme Oxoiron(IV) Complex in Aqueous Solution

**DOI:** 10.3390/molecules24183236

**Published:** 2019-09-05

**Authors:** Balázs Kripli, Bernadett Sólyom, Gábor Speier, József Kaizer

**Affiliations:** Department of Chemistry, University of Pannonia, 8201 Veszprém, Hungary

**Keywords:** catalase activity, iron(IV)-oxo, hydrogen peroxide, oxidation, kinetic studies

## Abstract

Heme-type catalase is a class of oxidoreductase enzymes responsible for the biological defense against oxidative damage of cellular components caused by hydrogen peroxide, where metal-oxo species are proposed as reactive intermediates. To get more insight into the mechanism of this curious reaction a non-heme structural and functional model was carried out by the use of a mononuclear complex [Fe^II^(N4Py*)(CH_3_CN)](CF_3_SO_3_)_2_ (N4Py* = *N*,*N*-bis(2-pyridylmethyl)- 1,2-di(2-pyridyl)ethylamine) as a catalyst, where the possible reactive intermediates, high-valent Fe^IV^=O and Fe^III^–OOH are known and spectroscopically well characterized. The kinetics of the dismutation of H_2_O_2_ into O_2_ and H_2_O was investigated in buffered water, where the reactivity of the catalyst was markedly influenced by the pH, and it revealed Michaelis–Menten behavior with *K*_M_ = 1.39 M, *k*_cat_ = 33 s^−1^ and *k*_2_(*k*_cat_/*K*_M_) = 23.9 M^−1^s^−1^ at pH 9.5. A mononuclear [(N4Py)Fe^IV^=O]^2+^ as a possible intermediate was also prepared, and the pH dependence of its stability and reactivity in aqueous solution against H_2_O_2_ was also investigated. Based on detailed kinetic, and mechanistic studies (pH dependence, solvent isotope effect (SIE) of 6.2 and the saturation kinetics for the initial rates versus the H_2_O_2_ concentration with *K*_M_ = 18 mM) lead to the conclusion that the rate-determining step in these reactions above involves hydrogen-atom transfer between the iron-bound substrate and the Fe(IV)-oxo species.

## 1. Introduction

Superoxide dismutases (SODs), catalase-peroxidases (KatGs) and catalases are specialized oxidoreductase enzymes for the degradation of reactive oxygen species (ROS), e.g., hydrogen peroxide, hydroxyl and superoxide radicals to avoid their accumulation and prevent the oxidative damage of cellular components, that may lead to a number of diseases such as cancer, Alzheimer’s diseases and aging [1,2,3,4]. For example, the hydroxyl and/or hydroperoxyl radicals may cause lipid peroxidation, membrane damage, DNA oxidation and cell death [5,6]. As a fine coupling of SODs and catalases, the former enzymes catalyze the dismutation of superoxide into dioxygen (1-electon oxidation) and H_2_O_2_, whilst the latter enzymes eliminate the H_2_O_2_ via its decomposition by disproportionation into O_2_ (2-electron oxidation) and H_2_O, resulting in the optimal intracellular concentration of a H_2_O_2_ molecule [7,8,9], which acts as a second messenger in signal-transduction pathways. Otherwise, it is worth to note, that the therapeutic potential of H_2_O_2_ makes this molecule also a valuable target in cancer killing via chemo- and radiotherapy, and in stroke therapy [10,11,12].

Two main classes of catalase enzymes are known, an iron and manganese-containing proteins. Although both types of catalases exhibit high catalytic activities, there are significant differences, including the active sites and the catalytic mechanisms [13]. Monofunctional catalases (EC 1.11.1.6) are heme-containing enzymes, that catalyze the dismutation of hydrogen peroxide (2H_2_O_2_ = 2H_2_O + O_2_), where the catalytic mechanism is well-characterized with a high-valent oxoiron(IV) porphyrin π-cation radical, compound I, [(P^•+^)Fe^IV^=O]^+^ (P = porphyrinate dianion), being responsible for hydrogen peroxide oxidation [14,15,16]. Manganese catalases such as *Lactobacillus plantarum* [17,18], *Thermus thermophilus* [19,20], *Thermoleophilium album* [21] and *Pyrobaculum calidifontis VA1* [22] are found in several bacterial organisms, and possess a binuclear manganese center with a cycle between Mn(II)-Mn(II) and Mn(III)-Mn(III) states during turnover.

Synthetic compounds as biomimics of catalase enzymes may have potential biomedical application as therapeutic agents against oxidative stress. Besides the heme-type models, a great number of manganese, copper, ruthenium and non-heme iron complexes have been designed and studied as catalase models [23,24,25,26,27,28,29,30,31,32,33,34,35]. However, comparative studies between heme and non-heme models are scarce. The non-heme models are mainly binuclear complexes [27,28,29], only a small number of mononuclear iron compounds have been studied [12,36,37]. The direct dismutation of H_2_O_2_ with terminal and bridging oxo ligands has been described for only a few complexes of Fe, Cr, Mn, V and Ru [38,39,40,41,42]. Mononuclear oxoiron(IV) complexes are of interest from a bioinorganic viewpoint, since similar intermediates are frequently invoked as the active species in the active site of numerous proteins and in biomimetic iron-containing catalytic systems. Most of these results were obtained in organic solvent due to the lack of solubility or activity in aqueous solution. Due to the increasing importance of catalase activity, we have focused on the development of such a non-heme iron-containing system that shows catalase-like activity in aqueous solution. To get more insight into the mechanism of H_2_O_2_ dismutation the mononuclear complex [Fe^II^(N4Py*)(CH_3_CN)](CF_3_SO_3_)_2_ (1) (N4Py* = *N*,*N*-bis(2-pyridylmethyl)-1,2-di(2-pyridyl)ethylamine) was chosen as a catalyst, where the possible reactive intermediates high-valent Fe^IV^=O (2) and Fe^III^-OOH (3) are known and spectroscopically well characterized (Scheme 1) [43,44,45,46].

## 2. Results and Discussion

### 2.1. Catalase-Like Reactivity of [Fe^II^(N_4_Py*)(CH_3_CN)](CF_3_SO_3_)_2_ in Aqueous Solution

The catalase-like activity of the complex [Fe^II^(N_4_Py*)(CH_3_CN)](CF_3_SO_3_)_2_ to disproportionate H_2_O_2_ into H_2_O and O_2_ was investigated in aqueous solution at 20 °C by gasvolumetric measurements of evolved dioxygen. To gain further information on the mechanism of catalase activity of our iron complex, we first examined pH-dependence of catalase activity. It was reported that the coordination and dissociation of peroxides on metal-porphyrins are pH dependent reactions [47,48]. Moreover, they reported that the coordination is accelerated at a higher pH region and that the subsequent O–O bond cleavage leading to the formation of high-valent oxo-Fe(IV) or oxo-Fe(V) species is pH-independent (only at higher pH region, where the protonation of the distal oxygen in the peroxo-complex can be excluded) irreversible reaction. These results suggest that the coordination of peroxides is a crucial step for the formation of high-valent Fe species, and the mechanism of catalase activity involves the coordination of H_2_O_2_, which is considered to be pH-dependent as well. Therefore, we hypothesized that formation of reactive intermediate 2 is accelerated at pH 9.5 and catalase activity is increased as compared at pH 8. As shown in Figure 1, O_2_ production of 1 in 50 mM borate buffer (pH 9.5) was significantly higher than that in phosphate buffer (pH 8). V_in_ value under this condition was determined to be V_in_ = 1.13 × 10^−3^ Ms^−1^, which is approximately seven times higher than that at pH 8, and 8.5 times higher than that at pH 11. This indicates that the rate-determining step was faster at pH 9.5 than at pH 8, which may be explained by the higher concentration of the more nucleophilic HO_2_^–^.

The pH dependence of H_2_O_2_ dismutation was further studied between pH 7 and pH 11. It was found that the initial rate of the disproportionation of H_2_O_2_ increases with increasing pH and goes through a maximum. The pH profile of 1 exhibits a sharp optimum at pH ~9.5, whereas catalases in general exhibit a broad pH optimum extending from pH 5.6 to 8.5 [48]. In control experiments, in the absence of the complex, the pH of the solution did not change in the presence of H_2_O_2_, and no significant O_2_ volume was evolved. We believe that the activity is influenced by the protonation state of H_2_O_2_. Assuming that hydrogen peroxide is activated by a direct interaction with the Fe^IV^=O group of the complex, decomposition is expected to be favored by a high pH because of the larger concentration of the hydroperoxide anion (HOO^−^ is more nucleophilic than H_2_O_2_). On the other hand, at higher pH values, the complex may be destroyed by the formation of the mineral forms of iron or catalytically inactive, insoluble μ-oxo-diiron(III) species.

Detailed kinetic studies on the disproportionation of H_2_O_2_ were performed in aqueous solution (pH 9.5; 0.025 M Na_2_B_4_O_7_.10H_2_O/0.1 M HCl; I = 0.15 M KNO_3_) at 20 °C by volumetric measurements of evolved dioxygen. To determine the dependence of the rates on the substrate concentration, solutions of the complex [Fe^II^(N_4_Py*)(CH_3_CN)](CF_3_SO_3_)_2_ were treated with increasing amounts of H_2_O_2_ (1:400–5300). Plots of the amount of dioxygen evolved versus time at [1]_0_ constant, are shown in Figure 1a. The initial rates values were calculated from the maximum slope of the O_2_ versus time curves. Under this experimental condition, saturation kinetics was found for the initial rates (V_in_ = –d[H_2_O_2_]/dt) versus the H_2_O_2_ concentration (Figure 1b). An analysis of the data based on the Michaelis–Menten model (V_in_ = k_cat_[cat][S]_0_/(K_M_ + [S]_0_)), originally developed for enzyme kinetics, was applied. A nonlinear least square fit was applied to calculate the Michaelis–Menten parameters, where k_cat_ is the turnover number, K_M_ is the Michaelis constant, S is the substrate initial concentration and [cat] is the catalyst concentration. The results were K_M_ = 1.39 M, k_cat_ = 33 s^−1^ and k_2_(k_cat_/K_M_) = 23.9 M^−1^s^−1^. The data presented illustrate that the catalyst had a relatively high turnover number (k_cat_) but appeared to bind peroxide very badly. The K_M_ value was greater than the values for the natural enzymes from *Thermus thermophilus* (K_M_ = 0.083 M) [19,20], *Tricholoma album* (K_M_ = 0.015 M) [21] and *Lactobacillus plantarum* (K_M_ = 0.35 M) [17,18] indicating a lower affinity to the substrate. The k_cat_ value equaled 33 s^−1^, however, was 3–4 times magnitudes lower when compared to the natural enzymes *Thermus thermophilus* (k_cat_ = 2.6 × 10^5^ s^−1^), *Tricholoma album* (k_cat_ = 2.0 × 10^5^ s^−1^), *Lactobacillus plantarum* (k_cat_ = 2.6 × 10^4^ s^−1^) and the heme-containing catalases (k_cat_ = 4 × 10^7^ s^−1^). Despite this iron complex presents lower values of catalytic efficiency than other models (Table 1) [49,50,51,52], it must be emphasized that this value was obtained in water and in pH close to the natural, representing an advantage of the title complex with respect to most of the published models, whose studies have been conducted in organic solvent due to the lack of solubility or activity in aqueous solution.

### 2.2. Catalase-Like Reactivity Mediated by [(N4Py*)Fe^IV^=O](ClO_4_)_2_ in Aqueous Solution

Rohde and co-workers have shown that the independently prepared [(N4Py)Fe^IV^=O]^2+^ reacts rapidly with near-stoichiometric H_2_O_2_ resulting in dioxygen and [Fe^II^(N4Py)(CH_3_CN)]^2+^ in acetonitrile [54]. Later Browne and co-workers have found clear evidence for the reaction of Fe^III^-OOH with H_2_O_2_ in methanol [55]. In their case the oxoiron(IV) intermediate can also be formed by homolytic cleavage of the O–O bond of an Fe^III^–OOH, but the rate of its formation is much lower than the Fe^III^–OOH-mediated H_2_O_2_ disproportionation observed with high excess H_2_O_2_ under catalytic conditions. As a continuity of these studies, we attempted to directly investigate the reactivity of the possible intermediates (Fe^IV^=O, Fe^III^–OOH) during the catalase reaction in aqueous solution.

We have shown earlier that complex 1 forms very stable high valent oxoiron(IV) species (2) with PhIO in CH_3_CN (t_1/2_ = 233 h at R.T., λ_max_ = 705 nm, ε = 400 M^−1^cm^−1^) [43]. As a test of our oxoiron(IV) species we firstly investigated its reaction with excess H_2_O_2_ (75 equiv.) in acetonitrile at 10 °C, which resulted in the formation of a relatively stable transient purple species with a characteristic absorbance maximum at λ_max_ 535 nm (ε = 1100 M^−1^ cm^−1^; Figure 2a). It had a half-life of about 3 min even at 25 °C, but its decay can be remarkably enhanced by the addition of H_2_O into the Fe^III^–OOH-containing solution (CH_3_CN/H_2_O = 1:1) with a k_obs_ value of about 12.3 × 10^−3^ s^−1^ at 10 °C, resulting in the formation of 2 (Figure 2b). It is worth to note that at higher pH the decay was so fast, that we were not able to follow it. These results might suggest that a high-valent oxoiron(IV) species was one of the possible intermediates that may be responsible for the dismutation of H_2_O_2_ in aqueous solution.

In the iron-catalyzed oxidation of H_2_O_2_ with terminal oxidants four processes can be proposed as the rate-controlling step, namely the formation of Fe^III^–OOH or high-valent oxoiron(IV), or their reaction with the substrate (H_2_O_2_). To avoid this difficulty, and to get more insight into the mechanism of the H_2_O_2_ oxidation process we synthesized the oxoiron(IV) complex 2 by an in situ reaction of 1 with PhIO in acetonitrile, and investigated its stability and reactivity with H_2_O_2_ in a buffered H_2_O–CH_3_CN mixture (*v*/*v* = 1:1). In this way the role of the oxoiron(IV) species could be directly investigated. The UV-vis spectra of 2 in buffered solutions were almost identical to that observed in the acetonitrile. The observed blue shift on the λ_max_ values (from 705 to 697 nm) might be explained by the interaction (H-bridge) of the oxoiron(IV) with the H_2_O molecule(s).

The stability of 2 was found to depend significantly on the pH value of reaction solutions, in which 2 was stable at pH 7–8 (k_sd_ = 0.43 × 10^−3^ s^−1^, 0.64 × 10^−3^ s^−1^ with t_1/2_ = 180 and 150 min at pH 7 and 8 at 10 °C, respectively), but decayed at a fast rate with increasing pH at pH 9–11 (k_sd_ = 3.51 × 10^−3^ s^−1^, and 7.27 × 10^−3^ s^−1^, 23 × 10^−3^ s^−1^, 39 × 10^−3^ s^−1^ and 46 × 10^−3^ s^−1^ with t_1/2_ = 4, 3, 2, 1.7 and 1 min at pH 9, 9.5, 10, 10.5 and 11 at 10 °C, respectively; Figure 3). This is the second example that the stability of oxoiron(IV) complex is controlled by the pH of reaction solutions [56].

The pH dependence of the reactivity of 2 against H_2_O_2_ was also examined in the range pH 7–11 in a buffered H_2_O–MeCN mixture (*v*/*v* = 1:1) at 10 °C (Figure 3). Upon addition of 10 equiv. H_2_O_2_ to the solution of 2, the characteristic absorption band of 2 (λ_max_ = 697 nm) disappeared rapidly, and no formation of Fe^III^–OOH was observed. Pseudo-first-order fitting of the kinetic data allowed us to calculate k_obs_ values to be 2.96 × 10^−3^ s^−1^, 6.29 × 10^−3^ s^−1^, 37.9 × 10^−3^ s^−1^, 41.6 × 10^−3^ s^−1^, 60.3 × 10^−3^ s^−1^, 75.3 × 10^−3^ s^−1^ and 84 × 10^−3^ s^−1^ at pH 7, 8, 9, 9.5, 10, 10.5 and 11 at 10 °C, respectively.

The reactivity of 2 was found to depend significantly on the pH value of reaction solutions. The maximum rate of H_2_O_2_ dismutation, *k*’_obs_ (*k*’_obs_ = *k*_obs_ − *k*_sd_ from the −d[2]/dt = *k*_obs_[2] = (*k*_sd_ + *k’*_obs_)[2]) could be observed at pH 9, where the self decay process (*k*_sd_) could be neglected (Figure 4a). The increase of the *k*_obs_ at higher pH could be explained by the self decay of 2. Addition of 10 equiv. H_2_O_2_ at pH 10 resulted in a decrease in absorbance at λ_max_ = 697 nm concomitant with an increase at 490 nm within 40 s at 10 °C, and an isosbestic point obtained at approximately λ_max_ = 620 nm. This spectrum including a weak absorption band at 700 nm with a shoulder around 490 nm corresponded to the spectrum of [(N4Py*)Fe^III^-O-Fe^III^(N4Py*)]^4+^ (Figure 4b).

Detailed kinetic and mechanistic studies were carried out in buffered water/acetonitril mixture (*v*/*v* = 1:1) in pH 8, close to the natural at 10 °C, where the self decay process can be excluded. The reactivity of 2 was monitored by UV-vis spectroscopy and the rate of its rapid decomposition was measured at 697 nm (Figure 5a). Pseudo-first order fitting of the kinetic data allowed us to determine k_obs_ values. These results indicate a direct reaction between 2 and H_2_O_2_. In order to investigate the possible involvement of a hydrogen atom in the rate-determining step we investigated the reactivity of 2 with H_2_O_2_ in buffered MeCN/D_2_O/H_2_O (*v*/*v* = 1:0.75:0.25). Solutions of 2 in the presence of D_2_O at pH 8 were somewhat less reactive against H_2_O_2_, yielding a solvent kinetic isotope effect of 6.2. This value was significantly smaller than that was obtained for the H–D isotope effect for [Ru^IV^O(bpy)_2_(py)] at pH 2.3 (KIE = 22.1 ± 1.2), but almost identical with that was measured at pH 9.7 (KIE = 8 ± 2.9) at 25 °C [40]. The most straightforward interpretation of the proton dependence was that the pathways involve the acid-base pre-equilibrium of H_2_O_2_ (H_2_O_2_ = HO_2_^−^+ H^+^) and the concomitant rate-controlling hydrogen-atom-transfer (HAT) between the Fe^IV^=O species and the OH (or OD) group of H_2_O_2_ (D_2_O_2_) [57] forming a peroxyl radical.

To determine the dependence of the rates on the substrate concentration, solutions of the complex [(N_4_Py*)Fe^IV^=O](CF_3_SO_3_)_2_ were treated with increasing amounts of H_2_O_2_ (1:5–50). Under this experimental condition, saturation kinetics was found for the k_obs_ versus the H_2_O_2_ concentration (Figure 5b). At low H_2_O_2_ concentration, a k’ value of about 0.47 M^−1^s^−1^ was obtained at 10 °C (k^’^ = k_obs_/[H_2_O_2_] assuming a first order dependence). The reactivity of 2 was lower than that of [(N_4_Py)Fe^IV^=O]^2+^ (N4Py = N,N’-bis(2-pyridylmethyl)-N-bis(2-pyridyl)methylamine) in CH_3_CN (k’ value of 8 M^−1^s^−1^ at 25 °C), but significantly higher than that of [(tmc)(CH_3_CN)Fe^IV^=O]^2+^ (tmc = 1,4,8,11-tetramethyl-1,4,8,11-tetraazacyclotetradecane; k_2_ value of 0.035 ± 0.002 M^−1^s^−1^ at 25 °C) in CH_3_CN. Furthermore, a k’ value of 12.7 ± 1.3 M^−1^s^−1^ had been reported for the oxoruthenium(IV) complex [Ru^IV^O(bpy)_2_(py)] at 25 °C (H_2_O, pH 7.92) [40]. Based on literature data, it can be concluded that [(N_4_Py*)Fe^IV^=O]^2+^ is more reactive in O–H bond activation (H_2_O_2_) than in C–H bond activation (hydrocarbons) [46].

Substrates saturation behaviors implied a rapid equilibrium between the unbound substrate and the iron complex as a result of hydrogen bridge bond. Under conditions of high substrate concentration, the primary species in solution was the Fe^IV^O–H_2_O_2_ (Fe^IV^O–HO_2_^−^) complex. The rate of the reaction was dependent only on the decomposition of the Fe^IV^O–H_2_O_2_ (Fe^IV^O–HO_2_^−^) complex (r.d.s.) to the product and free precursor complex (Scheme 2) [40,57]. A nonlinear least square fit was applied to calculate the Michaelis–Menten parameters. The results were K_M_ = 0.018 M, k_cat_ = 0.014 s^−1^ and k_2_(k_cat_/K_M_) = 0.754 M^−1^s^−1^. An apparent K_M_ value for bovine liver catalase (BLC) was determined to be 0.093 M. By contrast, the K_M_ values of KatGs (catalase-peroxidase) were much lower (0.0042 M for SynKatG, 0.0025 M for MtbKatG and 0.0059 M for BpKatG, all at pH 7) [48], but was almost identical with the value for the natural enzyme from *Tricholoma album* (K_M_ = 0.015 M) indicating a high affinity to the substrate, appearing to bind to peroxide very strongly [21].

## 3. Materials and Methods

The N_4_Py* ligand, and its [Fe^II^(N_4_Py*)(CH_3_CN)](CF_3_SO_3_)_2_ (1) complex were prepared according to published procedures [31]. UV/Vis spectra were recorded with an Agilent 8453 diode-array spectrophotometer (Agilent Technologies, Hewlett-Packard-Strasse 8, Waldbronn, Germany) with quartz cells.

Catalytic reactions were carried out at 20 °C in a 30 cm^3^ reactor containing a stirring bar under air. In a typical experiment the appropriate aqueous solution (19 cm^3^ 0.1 M KH_2_PO_4_/0.1 M NaOH pH 7, 8; 0.025 M Na_2_B_4_O_7_.10H_2_O/0.1 M HCl pH 9, 9.5, 10; or 0.05 M NaHCO_3_/0.1 M KOH pH 10.5, 11 buffer and I = 0.15 M KNO_3_) was added to the complex dissolved in 1 cm^3^ DMF, and the flask was closed with a rubber septum. H_2_O_2_ was injected by syringe through the septum. The reactor was connected to a graduated burette filled with oil, and the evolved dioxygen was measured volumetrically at time intervals of 15 s. Initial rates were expressed as Ms^−1^ by taking the volume of the solution into account, and calculated from the maximum slope of the evolved dioxygen versus time.

Stoichiometric reactions were carried out under thermostated conditions at 10 °C in 1 cm quartz cuvettes. In a typical experiment [Fe^II^(N_4_Py*)(CH_3_CN)](CF_3_SO_3_)_2_ (1) (3 × 10^−3^ M) was dissolved in acetonitrile (1.0 cm^3^), then iodosobenzene (4.5 × 10^−3^ M) was added to the solution. The mixture was stirred for 50 min then excess iodosobenzene was removed by filtration. The acetonitril solution was than diluted with the appropriate buffered aqueous solution (1.0 cm^3^), and the decay of 2 was followed by monitoring the decrease in absorbance at 697 nm (*ε* = 400 M^−1^ cm^−1^) in the absence or in the presence of H_2_O_2_ under a pseudo-first order condition of excess H_2_O_2_.

## 4. Conclusions

It was found earlier that non-heme oxoiron(IV) complexes were able to carry out electrophilic transformations including O–H activation of H_2_O_2_ via homolytic O–H bond cleavage in acetonitrile as a functional catalase model. As a continuity of this study, efforts were made to work out a functional model in aqueous solution, close to the natural, where the postulated oxoiron(IV) intermediate behaved as an electrophilic oxidant. In summary, we reported one of the first examples of catalytic and stoichiometric H_2_O_2_ dismutation into O_2_ and H_2_O in aqueous solution mediated by electrophilic oxoiron(IV) intermediate, where the reactivity of 2 was markedly influenced by the pH. Based on detailed mechanistic studies on H_2_O_2_ oxidation that were investigated with in situ generated oxoiron(IV) species, plausible mechanisms were proposed, in which the H_2_O_2_ oxidation occurred by the HAT mechanism. To put together the stoichiometric and catalytic results it could be said that the highest catalytic activity of the H_2_O_2_ dismutation could be observed at pH 9.5, where the concentration of the more nucleophilic hydroperoxide anion (HOO^−^) was high, and the self-decay of the oxoiron(IV) intermediate could be neglected. These results were in good agreement with the electrophilic reactivity of oxoiron(IV) intermediates proposed for heme-type monoiron catalases, and might help us to understand the mechanism of the detoxification of H_2_O_2_ in biological systems.

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
