# Peer review of "Stability and Catalase-Like Activity of a Mononuclear Non-Heme Oxoiron(IV) Complex in Aqueous Solution"

_molecules, 2019, doi:10.3390/molecules24183236_

Round 1

Reviewer 1 Report

The work describes an study of the catalase activity of an iron complex with a pentadentate ligand. The kinetic analysis is of high technical quality and given the extensive interest raise for these compounds over the last decade one finds strange that an study like the one described in the manuscript, which I find quite significant, has not been addressed before.

I have minor questions for the authors;

a) I dont know  if there is a problem with my computer but all the figures in the manuscript appear cut in the basis. Please check.

b) The KIE is quite interesting. The authors propose two possibilities for its origin. However, the KIE is determined from the decay of the Fe(IV)O species by reaction with H2O2. Shouldn't this imply that the origin is the HAT? What is the KIE under catalytic regime?

c) Scheme 2 is confusing and some reactions seem strange, for example the forth reaction from top seems quite unlikely to me. Has someone proposed this scheme before? If so, please provide references. If not, please reconsider.

d) How do the authors understand the nature of the first intermediate Fe(IV)O·H2O2. Is this an outer sphere intermediate?

Author Response

All those changes suggested by the referees have been implemented. Mistakes, inaccuracies have been discussed in details and corrected. The authors are grateful for the highly constructive critics and remarks that helped to increase the value of our manuscript.

a) I dont know if there is a problem with my computer but all the figures in the manuscript appear cut in the basis. Please check.

Answer (a): Probably there was a problem with the computer, because I didn't observe any problem.

b) The KIE is quite interesting. The authors propose two possibilities for its origin. However, the KIE is determined from the decay of the Fe(IV)O species by reaction with H2O2. Shouldn't this imply that the origin is the HAT? What is the KIE under catalytic regime?

Answer (b): Yes you are right.

It was rewritten:

"The most straightforward interpretation of the proton dependence is that the pathways involve the acid-base preequilibrium of H2O2 (H2O2 = HO2¯+ H+) and the concomitant rate-controlling hydrogen-atom-transfer (HAT) between the FeIV=O species and the OH (or OD) group of H2O2 (D2O2) [57] forming a peroxyl radical."

We didn't investigated the KIE under catalytic regime, because in this case there are many processes which may be effected during formation and decay of catalytically active species (FeOOH, FeIVO....). It would be complicated to assign the effect.  

c) Scheme 2 is confusing and some reactions seem strange, for example the forth reaction from top seems quite unlikely to me. Has someone proposed this scheme before? If so, please provide references. If not, please reconsider.

Answer (c): Yes you are right. The reference is provided and included.

Substrates saturation behaviors implies a rapid equilibrium between unbound substrate and the iron complex as a result of hydrogen bridge bond. Under conditions of high substrate concentration, the primary species in solution is the FeIVO-H2O2 (FeIVO-HO2¯) complex. The rate of the reaction is dependent only on the decomposition of the FeIVO-H2O2 (FeIVO-HO2¯) complex (r.d.s.) to the product and free precursor complex (Scheme 2) [40, 57].

d) How do the authors understand the nature of the first intermediate Fe(IV)O·H2O2. Is this an outer sphere intermediate?

Answer (d):

Substrates saturation behaviors implies a rapid equilibrium between unbound substrate and the iron complex as a result of hydrogen bridge bond. Under conditions of high substrate concentration, the primary species in solution is the FeIVO-H2O2 (FeIVO-HO2¯) complex.

Reviewer 2 Report

The manuscript dealing with a catalase-like biomimetic catalyst is on the whole well written, clear and concise, and it should merit publication. However, some points could be checked by the Authors to further enhance the quality of their work:

Line 30-31 The definition of aging as a disease seems to me at least debatable. The definition of the old age as maturity or on the contrary as the most serious disease is a philosophical topic rather than a scientific one…

L 53 Replace the comma after the references with a semicolon;

L 74-76 Are the Authors sure that the O—O bond cleavage in the Compound zero analogue is pH-independent? A robust mass of experimental as well as theoretical data suggest that lowering of pH - through protonation of the distal oxygen in Compound zero facilitates the bond cleavage…

L 90 Please insert “of the” between “pH” and “solution”;

L 94 Please change “perhydroxyl” into “hydroperoxide”, here and further; please delete the closed parenthesis after the hydroperoxide anion structure; please change “larger” into “higher”;

L 111 and the following Please italicize the Linnaean names of the cited microorganisms and fungi;

Figure 1 Please check the bottom of the Figure, which is partially obliterated by its legend;

Table 1 Please in Entry 10 change “thermophiles” into “termophilus”;

L 135 Please insert “in” between “resulting” and “dioxygen”;

L 151 Please change “intermediate” into “intermediates”;

L 264 Please explain what does “O—H activation of H2O2” mean;

L 275 Please delete “a” in front of “oxoiron(IV)”.

Author Response

All those changes suggested by the referees have been implemented. Mistakes, inaccuracies have been discussed in details and corrected. The authors are grateful for the highly constructive critics and remarks that helped to increase the value of our manuscript.

L 74-76 Are the Authors sure that the O—O bond cleavage in the Compound zero analogue is pH-independent? A robust mass of experimental as well as theoretical data suggest that lowering of pH - through protonation of the distal oxygen in Compound zero facilitates the bond cleavage…

Answer (a): Yes you are right. The sentence is rewritten.

It was reported that the coordination and dissociation of peroxides on metal-porphyrins are pH dependent reactions [47,48]. Moreover, they reported that the coordination is accelerated at a higher pH region and that the subsequent O-O bond cleavage leading to the formation of high-valent oxo-Fe(IV) or oxo-Fe(V) species is pH-independent (only at higher pH region, where the protonation of the distal oxygen in the peroxo-complex can be excluded) irreversible reaction.

L 264 Please explain what does “O—H activation of H2O2” mean;

Answer (b): It was rewritten.

It was found earlier that nonheme oxoiron(IV) complexes are able to carry out electrophilic transformations including O-H activation of H2O2 via homolytic O-H bond cleavage in acetonitrile as a functional catalase model.
